# B cell activation involves nanoscale receptor reorganizations and inside-out signaling by Syk

Kathrin Kläsener[1,2,3†], Palash C Maity[1,2,3†], Elias Hobeika[1,2,3], Jianying Yang[1,2,3], Michael Reth[1,2,3]*

[1]BIOSS Centre for Biological Signalling Studies, Albert-Ludwigs-Universität Freiburg, Freiburg, Germany; [2]Department of Molecular Immunology, Biology III, Faculty of Biology, Albert-Ludwigs-Universität Freiburg, Freiburg, Germany; [3]Max Planck Institute for Immunobiology and Epigenetics, Freiburg, Germany

**Abstract** Binding of antigen to the B cell antigen receptor (BCR) initiates a multitude of events resulting in B cell activation. How the BCR becomes signaling-competent upon antigen binding is still a matter of controversy. Using a high-resolution proximity ligation assay (PLA) to monitor the conformation of the BCR and its interactions with co-receptors at a 10–20 nm resolution, we provide direct evidence for the opening of BCR dimers during B cell activation. We also show that upon binding Syk opens the receptor by an inside-out signaling mechanism that amplifies BCR signaling. Furthermore, we found that on resting B cells, the coreceptor CD19 is in close proximity with the IgD-BCR and on activated B cells with the IgM-BCR, indicating nanoscale reorganization of receptor clusters during B cell activation.

*For correspondence: michael.reth@bioss.uni-freiburg.de

†These authors contributed equally to this work

Competing interests: The authors declare that no competing interests exist.

## Introduction

Engagement of the BCR results in proliferation of B cells and their differentiation into either antibody-secreting plasma cells or memory B cells (*Rajewsky, 1996*). In its monomeric form, the BCR is a 1:1 complex between the membrane-bound immunoglobulin (mIg) molecule and the Igα/Igβ-heterodimer (*Schamel and Reth, 2000*; *Tolar et al., 2005*). The cytoplasmic tails of both Igα and Igβ carry an immunoreceptor tyrosine-based activation motif (ITAM), the tyrosines of which are phosphorylated by Src family protein tyrosine kinases such as Lyn and the spleen tyrosine kinase (Syk) (*Schmitz et al., 1996*; *Pao et al., 1998*). However, the two kinases interact with the BCR in different ways. In contrast to Lyn, which predominantly phosphorylates only the first ITAM tyrosine, Syk phosphorylates both tyrosines and subsequently binds the double-phosphorylated ITAM (ppITAM) via its tandem SH2 domains (*Futterer et al., 1998*). The binding of Syk to a ppITAM sequence results in the rapid phosphorylation of ITAM tyrosines of neighboring receptors. The resulting increased Syk recruitment generates a positive feedback that amplifies signal transduction from the BCR (*Rolli et al., 2002*; *Mukherjee et al., 2013*).

A PLA study showed that Syk interacts only with the activated but not with the resting BCR (*Infantino et al., 2010*). How the ITAM sequence is shielded from the action of this kinase in resting B cells is not known but may involve cytoskeletal elements. Indeed, B cells can be activated not only by exposure to antigen but also to the F-actin inhibitor Latrunculin A (Lat-A), suggesting that the actin cytoskeleton protects the resting BCR (*Treanor et al., 2010*). The cytoskeleton is also limiting the free diffusion of receptors (*Kusumi et al., 2005*). In line with this, B cell activation is accompanied by cytoskeleton remodeling and increased BCR mobility. Furthermore, it has been shown that BCR mobility is reduced in membrane areas enriched in ezrin-radixin-moesin (ERM) proteins (*Harwood and Batista, 2011*).

**eLife digest** Our immune system protects us against diseases by recognizing invading pathogens, such as bacteria and viruses, and launching a response to eliminate them. In vertebrates, like mice and humans, this immune response often involves white blood cells called B cells, which make antibodies.

B cells can recognize a huge number of different molecules called antigens, including those from pathogens, with the help of their antigen receptors. These receptors are proteins that span the surface membrane of the B cells, such that most the receptor is outside of the cell, with the rest being inside the cell. When an antigen binds to the outside portion of a B cell receptor, that B cell becomes activated. The B cell then starts to multiply, and to produce antibodies that bind to that antigen and hence mark a pathogen for attack by the immune system.

For many years it was thought that two copies of the receptors had to be brought together for the B cell antigen receptor to activate the B cell. However, other research revealed that the receptors tend to cluster together in the membrane, even before an antigen is recognized. Now, Kläsener, Maity et al. have used techniques that can essentially measure the distance between two B cell antigen receptors, even when they are just a few billionths of a meter (or nanometers) apart. This revealed that the receptors start very close together, and actually move further away from each other when the B cells are activated.

Kläsener, Maity et al. also found that an enzyme, called spleen tyrosine kinase (Syk), is needed to separate the receptors. Further experiments revealed that Syk does this by binding to the so-called 'signaling motif' of the receptors, which is inside the cell: this causes the receptors to change shape, forcing the parts outside the cell to move apart. Furthermore, Kläsener, Maity et al. found that other proteins in the surface membrane called co-receptors—which cooperate with the B cell antigen receptors to activate a B cell—were also re-organized when B cells became activated.

It is likely that most other membrane proteins are also organized in clusters that are only nanometers across. As such, the techniques described by Kläsener, Maity et al. will now allow the study of membrane organization at the nanoscale; which, as yet, has remained largely unexplored.

However, and complicating matters further, the cytoskeleton not only plays an inhibitory, but also an activating role during B cell activation, as it is required for microcluster formation and receptor internalization (*Brown and Song, 2001*; *Cheng et al., 2001*).

Mature B lymphocytes coexpress an IgM-BCR and an IgD-BCR on their surface, the latter being the dominant receptors on these cells. These two receptors have distinct signaling functions and diffusion behaviors on the B cell surface but the molecular basis for these differences is poorly understood (*Kim and Reth, 1995*; *Treanor et al., 2010*). The cross-linking model (CLM) assumes that most of the 100,000 BCR complexes on the surface of a B cell are monomeric and that cross-linking of two BCR monomers by a polyvalent antigen initiates the signaling process (*Metzger, 1992*). Similarly, the conformation-induced oligomerization model (CIOM) proposes that antigen binding induces a conformational change of the monomeric BCR, resulting in the dimerization of the membrane proximal CH domain of the BCR followed by receptor signaling (*Tolar and Pierce, 2010*). However, recent studies show that antigen receptors on resting T and B cells are not uniformly distributed but are rather organized in nanoclusters inside protein islands (*Lillemeier et al., 2006*, *2010*; *Mattila et al., 2013*). In earlier biochemical studies employing the blue-native polyacrylamide gel electrophoresis (BN-PAGE) technique, we provided the first evidence for the oligomeric organization of the BCR (*Schamel and Reth, 2000*). Furthermore, with a quantitative bifluorescence complementation (BiFC) assay, we found that the resting BCR forms stable dimers on the cell surface while a BCR mutant that cannot form dimers is hyperactive and cannot be stably expressed on the B cell surface (*Yang and Reth, 2010b*). We thus have proposed the dissociation-activation model (DAM) whereby BCR activation is initiated by the dissociation of signaling-inert BCR oligomers (*Yang and Reth, 2010a*). We here explore the nanoscale organization of the BCR at a 10–20 nm resolution with a modified version of PLA. With this technique, we have found direct evidence for the opening of BCR dimers during B cell activation and show that the kinase Syk plays an essential role in this process.

## Results

### Fab-PLA: a robust assay to test for BCR opening and reorganization

BCR monomers or dimers have sizes in the 10–20 nm range and thus, these forms cannot be distinguished by live cell imaging techniques with a diffraction barrier of 250 nm (*Huang et al., 2010*). To analyze the BCR conformation in the nanoscale range, we employ different versions of the PLA method using DNA-oligo-coupled antibodies, a rolling circle amplification and detection by fluorescence-coupled oligonucleotides (*Soderberg et al., 2008*). The classical PLA method (2-PLA) involves secondary antibodies and detects the proximity of two molecules in the 10–80 nm range. By coupling the DNA oligos directly to primary antibodies (1-PLA), or to Fab fragments (Fab-PLA), we narrowed the detection range of this assay down to 10–40 nm and 10–20 nm, respectively (*Figure 1A–C*). We then used these three PLA assays, and a monoclonal antibody directed against the constant part of IgM, to analyze the BCR organization on resting or activated TKO-MD B cells stimulated for 5 min with antigen (*Figure 1D–F*). TKO-MD is a pro-B cell line derived from RAG, Lambda5, SLP-65 triple-deficient mouse bone marrow which is transfected to express an IgM- and IgD-BCR specific for the hapten 4-hydroxy-5-iodo-3-nitrophenyl acetyl (NIP) (*Meixlsperger et al., 2007*). As indicated by the numbers of fluorescent dots, the 2-PLA and 1-PLA are more sensitive than the Fab-PLA, a phenomenon that is explained by the avidity associated with divalent antibody binding. However, only the Fab-PLA showed a clear difference in the relative IgM:IgM conformation or distance between resting and activated B cells (*Figure 1F*). Importantly, we are using non-permeabilized, fixed B cells in these assays. This allows us to analyze the organization of molecules specifically on the cell surface without detecting intracellular complexes of these molecules.

The PLA data were quantified by the BlobFinder software (*Allalou and Wahlby, 2009*) and showed that the Fab-PLA differences between resting and activated B cells are statistically significant (*Figure 1G*). PLA is a sampling method that, by measuring a few interactions per cell, allows conclusions to be drawn about the behavior of a larger pool of molecules. If we increase the Fab concentration, we can measure many more IgM:IgM interactions on resting but not on activated B cells (*Figure 1H,I*). However, at these concentrations, the signals (red blobs) are often overlapping, making it more difficult to quantify. We therefore use a concentration of the oligo-labeled Fab fragment that detects 5–10 interactions on each cell and collect data from hundreds of B cells in each experiment. All experiments are then repeated 3–5 times.

The IgM–IgM PLA studies suggests that, upon B cell activation, mIgM molecules move further than 20 nm apart from each other so that their proximity can be still detected by 1-PLA but no longer by Fab-PLA (*Figure 1J*). To provide further proof that the Fab-PLA method is able to distinguish oligomeric from monomeric IgM, we studied different forms of soluble IgM bound to NIP-coupled latex beads (*Figure 2A*). For this we purified monomeric and pentameric IgM from the supernatant of the anti-NIP specific hybridoma cell line B1-8 (*Figure 2B*). NIP-coupled latex beads were exposed to low amounts of either monomeric or pentameric IgM and equal IgM loading was determined by FACScan using anti-μ fluorescent antibodies (*Figure 2C*, lower left panel). The beads were then subjected to Fab-PLA and 1-PLA. The FACScan analysis of these beads shows that Fab-PLA detects the pentameric IgM but not the loosely spaced monomeric IgM whereas 1-PLA detects both forms of IgM equally well (*Figure 2C*, compare upper and lower panels). This result clearly shows that the Fab-PLA specifically detects oligomeric forms of IgM be it the soluble IgM pentamer or the oligomeric IgM-BCR on the B cell surface. In a recent structural study of pentameric IgM, it was shown that the constant domains of IgM are no further apart from each other than 5–6 nm (*Czajkowsky and Shao, 2009*), a distance that can be easily bridged by the anti-IgM Fabs used for the Fab-PLA method.

To exclude that the loss of the PLA signal in activated B cells is due to a loss of binding of the Fab fragment rather than the oligomer to monomer transition we measured the binding of all our Fab reagents by flow cytometry and found no alteration of binding between resting and activated B cells (*Figure 2—figure supplement 1*). In addition we used the same anti-Ig Fabs to monitore the heavy (H) and light (L) chain interaction by H:L Fab-PLA and found, as expected, no alteration upon B cell activation. In contrast to the H:L, the IgM:IgM and IgD:IgD (see below) Fab-PLA signal is consistently lost in activated B cells suggesting that the later assays detect dynamics states of a protein complex and we think that this is the oligomer to monomer transition of the BCR. In case of the IgD-BCR, we found direct evidence for this notion, because the IgD:IgD Fab-PLA detects the wild-type receptor but barely detects a mutant IgD-BCR that is defective in oligomer formation (*Figure 3*). For the latter assay, we

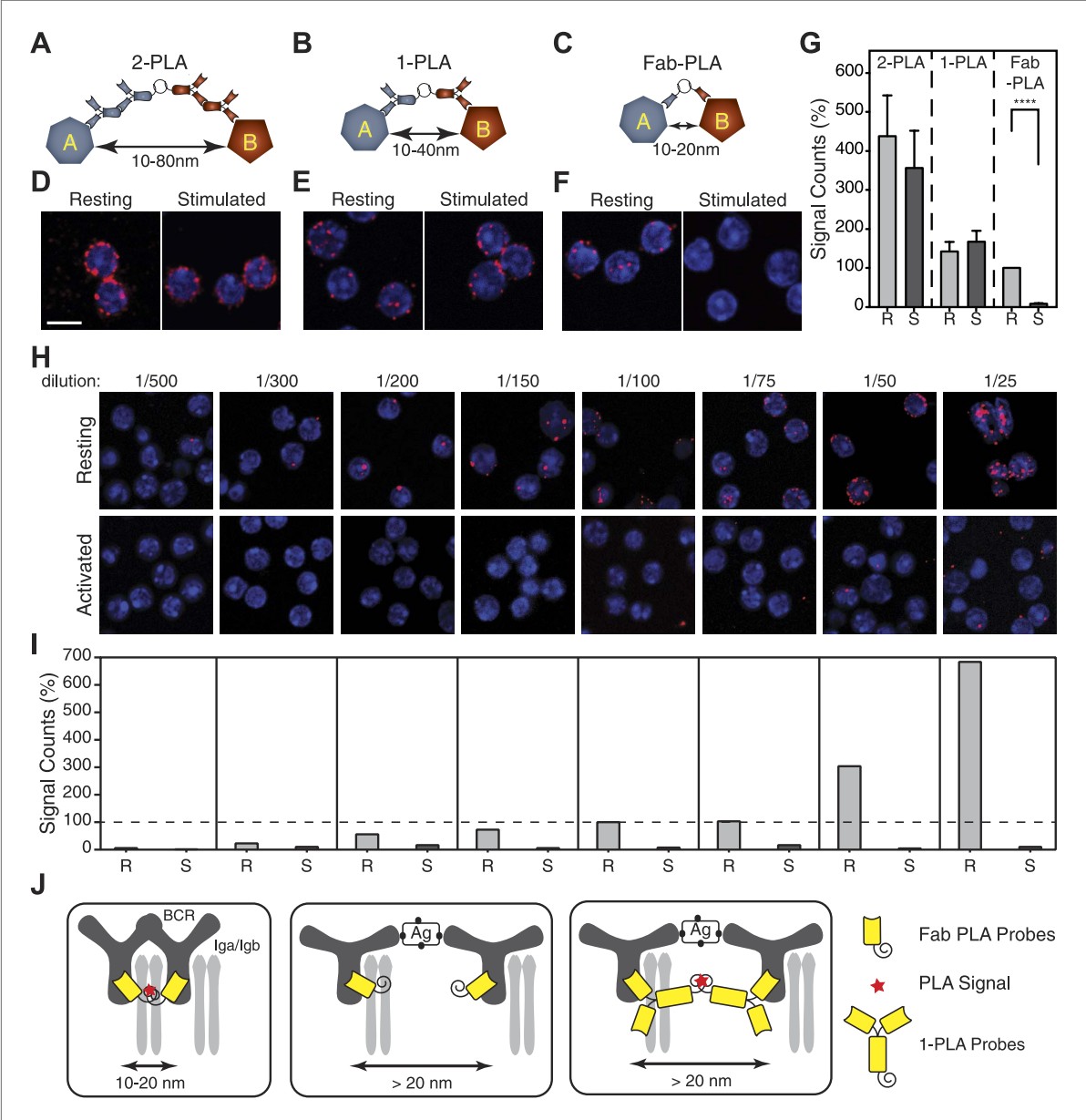

**Figure 1**. Nanoscale analysis of the IgM:IgM BCR proximity on the surface of TKO-MD B cells by PLA of different settings. (**A**–**C**) Scheme of the three PLA settings. Plus and minus oligos for annealing and rolling circle amplification were coupled to either (**A**) secondary antibodies (2-PLA), (**B**) primary antibodies (1-PLA) or (**C**) Fab fragments (Fab-PLA). The theoretical detection ranges are noted under the double-heads-arrows for each setting. (**D**–**F**) Representative confocal microscopic images of IgM:IgM PLA of resting (left) and antigen (NIP-BSA) activated (right) TKO-MD cells by (**D**) 2-PLA, (**E**) 1-PLA or (**F**) Fab-PLA. Nuclei were stained with DAPI and presented as blue signals while the PLA signals are presented as red color dots. Scale bar: 5 µm. (**G**) Quantification of the results of IgM:IgM BCR proximity for resting (R) and NIP-BSA-stimulated (S) TKO-MD cells analyzed by 2-PLA, 1-PLA and Fab-PLA. For each experiment, the PLA signals (counts per cell) of each sample were counted from a minimum of 50 cells and then normalized to the PLA signal of the resting cells with Fab-PLA (set to 100%). Data represent the mean and SEM of five independent experiments. Significant difference between samples is highlighted by stars. (**H** and **I**) Representative confocal microscopic images (**H**) and quantified results (**I**) of IgM:IgM Fab-PLA of resting and NIP-BSA stimulated TKO-MD cells using different concentrations of Fab reagents. Data represent the mean and SEM of a minimum of three independent experiments. For each experiment, the PLA signals (counts per cell) of each sample were counted from a minimum of 1000 cells and were then normalized to the PLA signal of the resting B cells probed with the 1/100 Fab dilution. (**J**) Schematic drawing showing the spatial organization of the BCR as monitored by Fab-PLA on resting and stimulated B cells (left and middle panel) or 1-PLA on stimulated B cells (right panel).

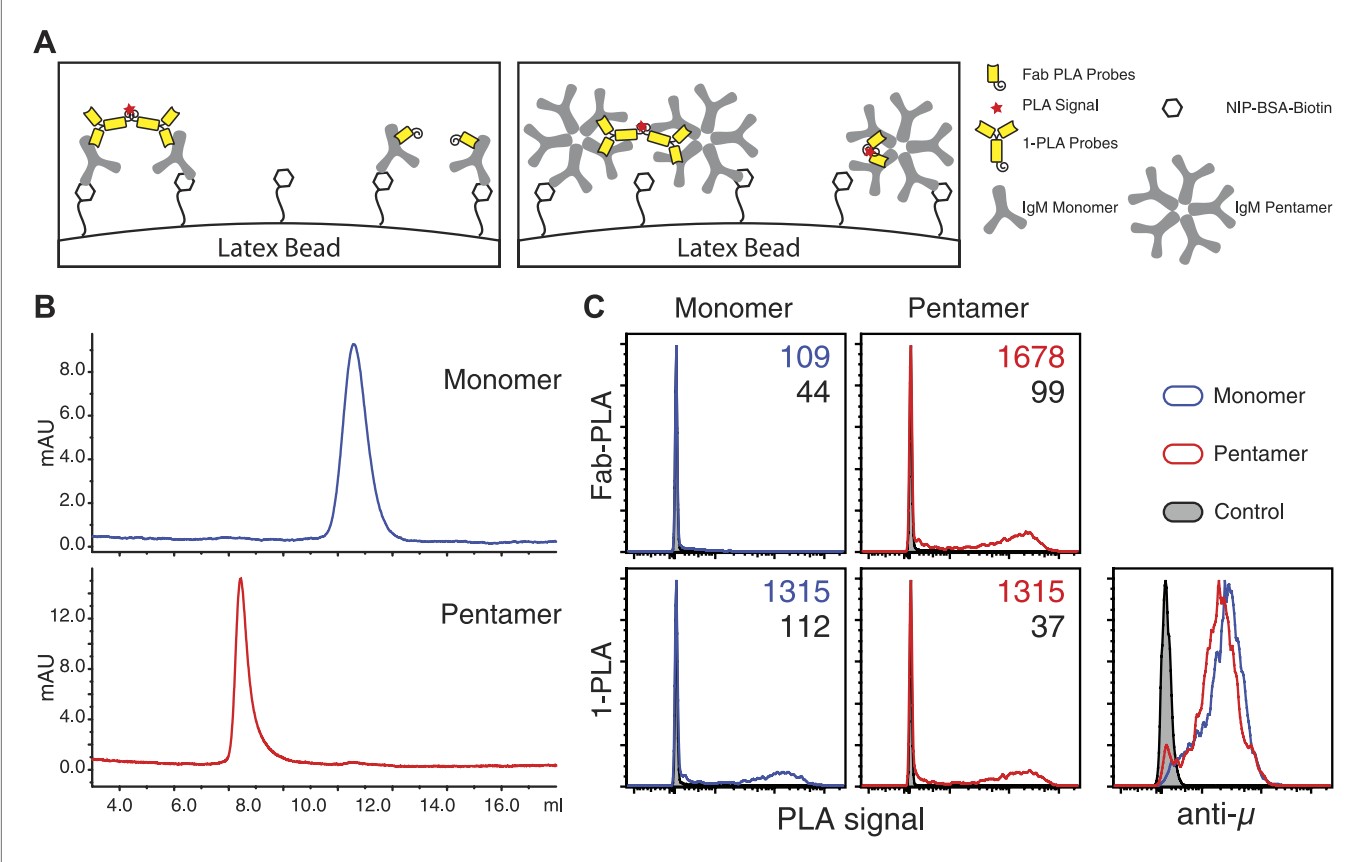

**Figure 2**. Fab-PLA but not 1-PLA is able to distinguish the oligomeric and monomeric BCR. (**A**) Scheme of the experiment setting. Latex beads coupled with low density of NIP-BSA were loaded with either the monomer or pentamer form of NIP-specific IgM antibodies. Due to the theoretical difference in their detection range (*Figure 1J*), 1-PLA is expected to obtain positive signals for beads loaded with either the monomeric or the pentameric IgM, while Fab-PLA is expected to show positive only for beads bound with pentameric IgM. (**B**) The size and purity of monomeric and pentameric IgM preparations were verified by size exclusion chromatography. (**C**) Comparison of beads loaded with monomeric (left) or pentameric (right) IgM measured by FACS after Fab-PLA (up) or 1-PLA (low) assay. Results were gated for the beads based on SSC and FSC. MFI of the PLA signals for the beads are marked on the upper right corner of each plot. The amount of IgM bound to the beads were monitored by anti-µ staining and shown at the lower right corner. Data are representative of three independent experiments.

The following figure supplements are available for figure 2:

**Figure supplement 1**. Similar binding efficiency of Fab-PLA probes on resting and stimulated B cells.

used transfected S2 cells because unlike B cells they allow the equal expression of wild-type and monomeric mutant IgD-BCR (*Yang and Reth, 2010b*).

We next used the Fab-PLA method to further analyze the IgM:IgM, and the IgD:IgD proximity on the surface of resting or activated spleen cells from the B1-8 mouse, whose B cells express a NIP-specific BCR (*Sonoda et al., 1997*). For this, we stimulated the B cells for 5 min with the antigen NIP-BSA (*Figure 4A*). The quantification of the Fab-PLA analysis of B1-8 (*Figure 4A*, right panel) and TKO-MD (*Figure 4B*) B cells shows that in each case, the IgM:IgM and IgD:IgD Fab-PLA signal is lost upon exposure of B cells to antigen. The Fab-PLA signal is also lost when we stimulated B1-8 B cells apart from antigen with BCR-activating drugs such as pervanadate (Perv) and Lat-A (*Figure 4C*), which alter the phosphorylation/dephosphorylation equilibrium and inhibit actin polymerization, respectively (*Secrist et al., 1993*; *Treanor et al., 2010*). The same results were also obtained by an analysis of human peripheral blood B cells (*Figure 4D*). Together, these studies suggest that B cell activation is accompanied by the opening of tightly packed BCR oligomers as predicted by the DAM hypothesis (*Yang and Reth, 2010a*, *2010b*).

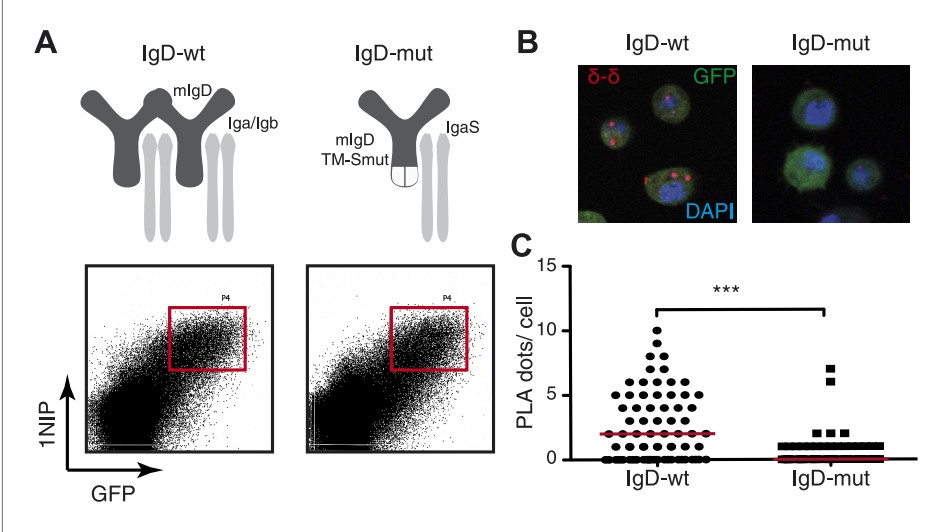

**Figure 3**. Fab-PLA detects IgD-BCR oligomers on the S2 cell surface. (**A**) Schematic drawing of wild type and double-mutant IgD-BCRs (δm transmembrane mutations and removal of the S–S bridge of the Igα/Igβ heterodimer) and their expression on transfected S2 cells analyzed by FACScan with a fluorescent 1NIP-peptide. Positively transfected S2 cells are indicated by the expression of a cotransfected GFP vector. The double-positive (GFP+, BCR+) S2 cell population, indicated by red square, were sorted and used for Fab-PLA. (**B**) Confocal microscopy analysis of the IgD:IgD Fab-PLA reaction of GFP+ S2 cells expressing wild type (left panel) or double-mutant (right panel) IgD-BCR. PLA signals are shown as red dots and nuclei were visualized by DAPI staining (blue). Scale bar: 5 μm. (**C**) Quantification of IgD:IgD Fab-PLA (each dot represents the amounts of Fab-PLA signals per S2 cell). The data were analyzed by the mann-whitney test and the median values are shown as red line.

## Syk opens the BCR oligomer by an inside-out signaling mechanism

The above analysis shows that BCR opening does not always require the direct engagement of the receptor but can also be mediated by drugs such as Lat-A and pervanadate which alter the cytoskeleton or kinase/phosphatase equilibrium, respectively. These treatments also result in strong ITAM phosphorylation. We thus wondered whether the detected BCR opening is associated with the Syk/ITAM signal amplification process we have described previously (*Rolli et al., 2002*) and that was recently confirmed by another study (*Mukherjee et al., 2013*). To test this, we pretreated splenic B cells with the Src-family kinase inhibitor PP2, or the Syk inhibitor R406, prior to their activation with Lat-A (*Braselmann et al., 2006*). While the PP2-treatment only delays the opening of the IgM-BCR oligomer, this process no longer occurs in the presence of the Syk inhibitor R406 (*Figure 5A,B*). The dependence on Syk activity was also found for B cells stimulated with the antigen NIP-BSA (*Figure 5— figure supplement 1*). To exclude that the block in BCR opening by PP2 or R406 is not due to off-target effects of these inhibitors, we analyzed the BCR conformation in Lyn- or Syk-deficient splenic B cells before and after their activation with Lat-A (*Figure 5C*). The IgM:IgM Fab-PLA results of these genetically altered B cells were similar to those previously observed using kinase inhibitors and suggests that Syk not only signals downstream but also alters the BCR conformation via an inside-out signaling process.

An inside-out signaling mechanism has previously been described for integrins such as LFA-1, where the binding of talin to the intracellular tails changes the outside conformation and binding behavior of this receptor (*Wang, 2012*). Accordingly, it may be that Syk binding to the phosphorylated ITAM tyrosines induces receptor opening. We therefore used Igα:Syk Fab-PLA to analyze whether or not the kinase inhibitors PP2 and R406 influence the binding of Syk to the BCR in Lat-A-stimulated B cells. For this study, the B cells were fixed and permeabilized to allow probing of the inner surface of the plasma membrane. This analysis shows that the BCR/Syk interaction is induced within 1 min of Lat-A treatment (*Figure 5D,E*). This interaction is delayed by PP2 whereas the Syk inhibitor R406 completely blocks the recruitment of Syk to the BCR. The inside-out signaling function of Syk is thus likely to be mediated not only by its kinase activity, but also by the binding of Syk to the ITAM sequences of

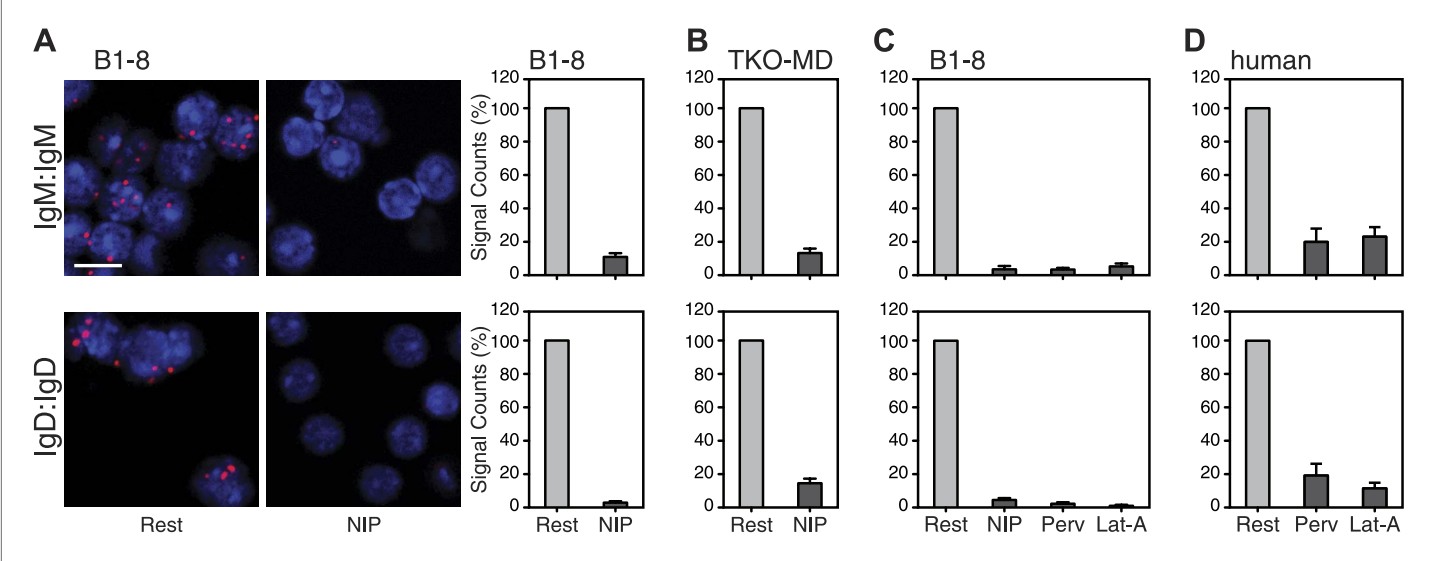

**Figure 4**. Fab-PLA study of the nanoscale organization of the IgM-BCR and the IgD-BCR on resting and activated B cells. (**A**) The IgM:IgM (upper) and IgD:IgD (lower) proximity of antigen receptors on resting or activated B1-8 splenic B cells were examined by Fab-PLA and shown as representative microscopic images (left) and quantified results (right). (**B**–**D**) Quantified Fab-PLA results indicate the IgM:IgM (upper) and IgD:IgD (lower) proximity on resting or activated TKO-MD cells (**B**), B1-8 splenic B cells (**C**) and human IgM+IgD+ naïve B cells isolated from peripheral blood (**D**). Scale bar: 5 μm. Quantified data represent the mean and SEM of a minimum of four independent experiments. For each experiment, PLA signals (counts per cell) of each sample were counted from a minimum of 100 cells and were then normalized to the PLA signals of the resting cells.

the BCR. To again exclude off-target effects of inhibitors, we studied the ITAM/Syk association also in Lyn- and Syk-deficient splenic B cells before or after their activation with Lat-A (***Figure 5F***). Note that the ITAM/Syk interaction is delayed but not prevented in Lyn-deficient B cells, whereas the Igα:Syk Fab-PLA in Syk-deficient B cells gave no signal due to the absence of Syk.

The Syk/BCR inside-out signaling process, described here for the first time, may allow B cells to become fully active when exposed to low amounts of antigens. If this were true, one would predict that under conditions of Syk inhibition, B cells would require more antigen for the opening of the many IgM-BCR oligomer complexes on their cell surface. To test for this, we exposed TKO-M cells for different times to the Syk inhibitor R406 (5 μM) and then stimulated these B cells for 5 min with increasing doses of antigen (***Figure 5G***). Note that in the cytosol there exists an equilibrium between a closed autoinhibited and an open active form of Syk. As the inhibitor is interacting only with the latter form, it takes some time for the majority of Syk molecules in the cell to be blocked by the inhibitor. Syk-sufficient cells require 10 ng/ml of antigen to lose the IgM:IgM Fab-PLA signal whereas B cell exposed for 60 min to the Syk inhibitor lose the IgM:IgM Fab-PLA signal only at 10–50 times higher (250–500 ng/ml) antigen doses. This titration experiment supports the notion that the Syk/BCR inside-out signaling increases the sensitivity of B cells to low doses of antigen. B cell activation may thus involve two distinct phases (***Figure 5H***). First an outside-in signal where limited amounts of antigens open a few BCR dimers, allowing them to form active BCR/Syk seed complexes that then diffuse in the membrane and phosphorylate neighboring BCR oligomers. This process then results in more Syk recruitment, BCR opening and the amplification of the BCR signal via an inside-out signaling mechanism.

To show the influence of Syk on the BCR conformation more directly, we performed a gain-of-function experiment in the S2 Schneider cell system, which allowed us to express the BCR, either alone or together with its signaling components (***Yang and Reth, 2012***). On IgM-BCR-expressing S2 cells, we detect receptor dimers by IgM:IgM Fab-PLA (***Figure 6A,B***). The IgM:IgM Fab-PLA signal is drastically reduced on S2 cells co-expressing the IgM-BCR together with a GFP-Syk fusion protein (***Figure 6A***, second panel and ***Figure 6B***, second bar). The opening of the BCR by Syk could be either due to the phosphorylation of the ITAM tyrosines or to the binding of the tandem SH2 domains to the ppITAM sequences. To distinguish between these possibilities, we separated the enzymatic and binding domain of Syk from each other. For this, we expressed in S2 cells the chimeric Syk protein

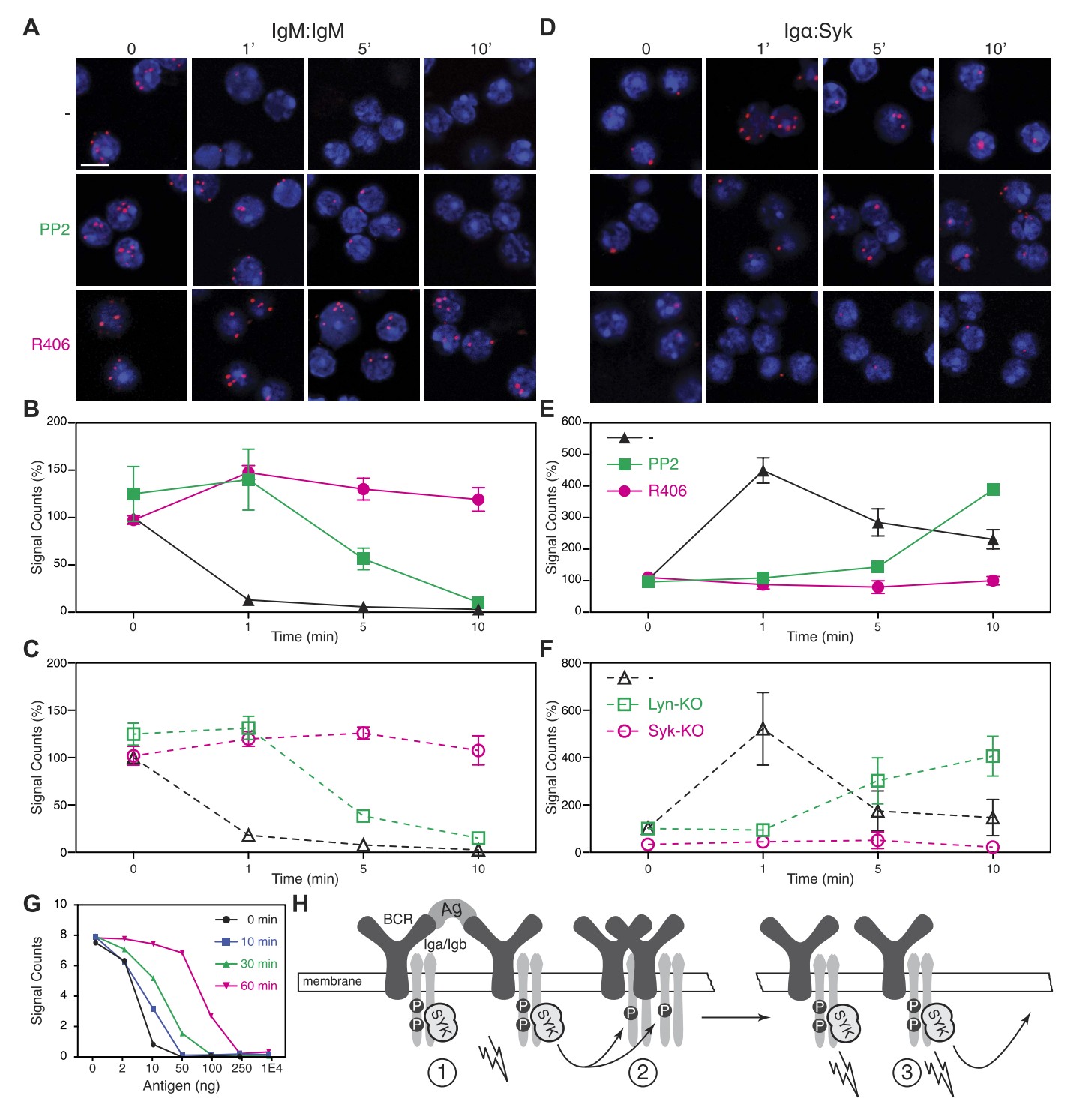

**Figure 5**. Syk activity is required for an efficient IgM-BCR dissociation after B cell activation. Representative microscopic images of the IgM:IgM (**A**) and the Igα:Syk (**D**) proximity detected by Fab-PLA. B1-8 B cells were analyzed at different time points (0, 1, 5, 10 min) after their activation with Lat-A in the absence (upper panels) or presence of the src-family kinase inhibitor PP2 (middle panels) or the Syk inhibitor R406 (bottom panels). Scale bar: 5 μm. Quantification of the IgM:IgM (**B**) and the Igα:Syk (**E**) Fab-PLA plotted as mean and SEM of a minimum of three independent experiments. For each experiment, PLA signals (counts per cell) of each sample were counted from a minimum of 500 cells and were then normalized to the PLA signals of resting untreated B1-8 B cells. Quantification of the IgM:IgM (**C**) and the Igα:Syk (**F**) proximity detected by Fab-PLA. B cells isolated from spleens of wild type, Lyn-KO and inducible, B cell-specific Syk-KO mice were analyzed at different time points (0, 1, 5, 10 min) after their activation with Lat-A. For each

*Figure 5. Continued on next page*

*Figure 5. Continued*

experiment, PLA signals (counts per cell) of each sample were counted from a minimum of 500 cells and were then normalized to the PLA signals of resting untreated wild type B cells. Data represent the mean and SEM of a minimum of three independent experiments. (**G**), Quantification of the IgM:IgM Fab-PLA of TKO-M cells stimulated for 5 min with the indicated amounts (2, 10, 50, 100, 250, 10000 ng/ml) of the antigen NIP-BSA at different times (0, 10, 30, 60 min) after the pretreatment of the B cells with the Syk inhibitor R406. PLA signals (signal counts) of each sample were counted from a minimum of 500 cells. Data are representative of three independent experiments. (**H**) Schematic drawing showing the spreading and amplification of the BCR/Syk signal by an outside-in and inside-out signaling mechanism. (1) Formation of the BCR/Syk seed complex by antigen-engaged BCRs; (2) ITAM phosphorylation of neighboring (unengaged) BCR complexes; (3) Binding of Syk to the phosphorylated ITAM opens the receptor from the inside (inside-out signaling) and results in further signal spreading.

The following figure supplements are available for figure 5:

**Figure supplement 1**. Syk activity is required for an efficient IgM-BCR dissociation on antigen stimulated B cells.

nLyn-Syk-KD consisting of the N-terminal myristoylation anchor of Lyn and the isolated Syk kinase domain either alone or in combination with a GFP-(SH2)2 construct containing the tandem SH2 domains of Syk. S2 cells expressing the IgM-BCR only together with nLyn-Syk-KD show tyrosine phosphorylation of Igα (data not shown) but no loss of the IgM:IgM Fab-PLA signal (***Figure 6A***, third panel

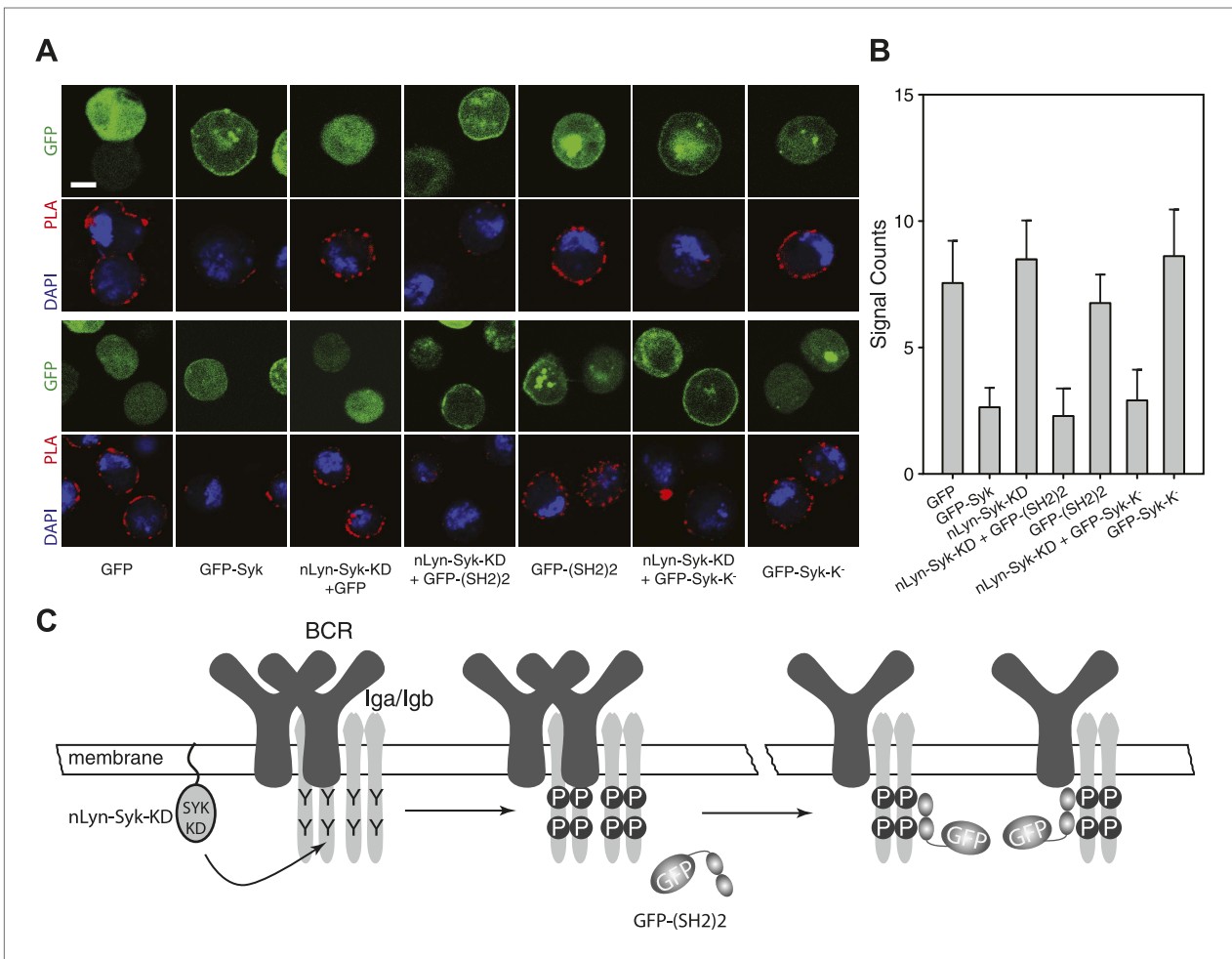

**Figure 6**. SH2 domains of Syk bind to phosphorylated ITAM and open the BCR. Representative microscopic images (**A**) and quantified results (**B**) show the IgM:IgM proximity of S2 cells expressing IgM-BCR together with the indicated constructs. GFP is shown as green, nuclei are stained with DAPI and shown as blue signals. The PLA signal is indicated by the red color. Scale bar: 5 μm. For the quantification, PLA signals (signal counts) of each sample were counted from a minimum of 500 cells. (**C**) Schematic drawing showing that not the ITAM phosphorylation tyrosines but the binding of the two tandem SH2 domains of Syk to the phosphorylated ITAM tyrosines is opening the BCR.

and *Figure 6B*, third bar). However, the oligomeric IgM-BCR opened when we co-expressed nLyn-Syk-KD together with either the GFP-(SH2)2 or the kinase negative GFP-Syk-K⁻ constructs that alone did not alter the IgM-BCR conformation (compare relevant panels and bars in *Figure 6A,B*). Together these data indicate that it is not the phosphorylation of the ITAM tyrosines but rather the binding of the Syk to the receptor that results in the opening of BCR oligomers (*Figure 6C*).

## Class-specific association of the BCR with the co-receptors CD19 and CD20

Signaling from the BCR is controlled and further amplified by BCR co-receptors, most prominently by the CD19 molecule (*Sato et al., 1995*; *Fujimoto et al., 2000*). CD19-deficient B cells have a compromised immune response to antigen and are defective in BCR microcluster formation (*Rickert et al., 1995*; *Depoil et al., 2008*). We therefore studied the IgM:CD19 and IgD:CD19 organization on resting and activated TKO-MD B cells by Fab-PLA (*Figure 7A,B*). This analysis shows that CD19 is only found in close association with the activated but not with the resting IgM-BCR (*Figure 7A*). Interestingly, in this assay, the results for the IgD-BCR were opposite to those observed for IgM. CD19 is found together with the resting IgD-BCR and dissociates from this receptor upon BCR activation (*Figure 7B*). The quantification of these data shows, for the first time, that the CD19:BCR interaction is Ig class specific and remodeled upon B cell activation (*Figure 7A,B*, right panel). Compared to CD19, less is known about the function of CD20 on the B cell surface but this molecule plays an important role as the target of the therapeutic antibody rituximab (*Korhonen and Moilanen, 2010*). By IgM:CD20 and IgD:CD20 Fab-PLA, we found that CD20 displays the same class-specific BCR interaction as CD19 (*Figure 7A,B*). The BCR:co-receptor analysis described above for TKO-MD cells was repeated with human blood (*Figure 7C,D*) and murine splenic B1-8 (*Figure 7E,F*) B cells and showed the same class-specific co-receptor association.

BCR opening, as well as the reorganization of co-receptors, does not require the co-expression of the IgM and IgD on the cell surface. TKO-M and TKO-D cells that only express one class of BCR behave similarly to the TKO-MD cells (*Figure 7—figure supplement 1*). Resting TKO-D cells display a close proximity between the IgD-BCR and the CD19/CD20 module but lose this interaction upon BCR stimulation, while the IgM-BCR gains contact to the CD19/CD20 module only upon B cell activation. These results suggest that on resting B cells, the CD19/CD20 module resides in a membrane compartment that is also targeted by the IgD-BCR but that is not dependent on IgD expression. To learn more about the lipid composition of this compartment, we labeled the cholera toxin B-subunit (CTB) with one of the PLA oligos. CTB binds with high affinity to GM1 gangliosides, glycolipids that are highly expressed in lipid ordered domains (*Kenworthy et al., 2000*). The IgD:CTB and IgM:CTB PLA showed that on resting B cells, GM1 gangliosides are found in close proximity to the IgD-BCR, whereas the IgM-BCR gains access to large amounts of these lipids only upon B cell activation (*Figure 7H,G*). The GPI-linked protein CD52 (*Treumann et al., 1995*) showed a distribution by Fab-PLA similar to that seen for the GM1 gangliosides (*Figure 7—figure supplement 2*). Together, these studies suggest that the IgM-BCR and the IgD-BCR are localized on the B cell surface in different membrane areas that are reorganized upon B cell activation.

## Discussion

The Fab-PLA method we have developed, makes it possible, for the first time, to explore the organization of receptors on the surface of normal B cells in the 10–20 nm range. Our analysis suggest that antigen-dependent B cell activation is accompanied by a transition from a tightly packed BCR oligomer to a more loosely spaced BCR-antigen or BCR-Syk clusters that can no longer be detected by Fab-PLA (*Figure 1J*, *Figure 5H*). Our findings support the DAM hypothesis (*Yang and Reth, 2010a*, *2010b*), and in addition show for the first time that the kinase Syk is necessary and sufficient for the opening of the oligomeric BCR. Importantly, all these observations could only be made with Fab-PLA but not with 1-PLA or the classical 2-PLA techniques, indicating that these receptor reorganization processes occur at 10–20 nm distances. The reason why only Fab-PLA can reliably monitor the alterations of the BCR conformation may be related to the structure of this receptor. Like an antibody, the BCR contains flexible Fab arms and it maybe the extension of these arms that moves the BCR monomers apart from each other so that Fab-PLA can no longer detect their proximity (*Figure 1J*). Another important advantage of Fab-PLA is that it does not require the overexpression of wild type or altered (tagged) proteins in cell lines but allows studying the nanoscale organization of unchanged proteins on primary cells. The Fab-PLA is a surprisingly robust technique that detects the same BCR alterations in

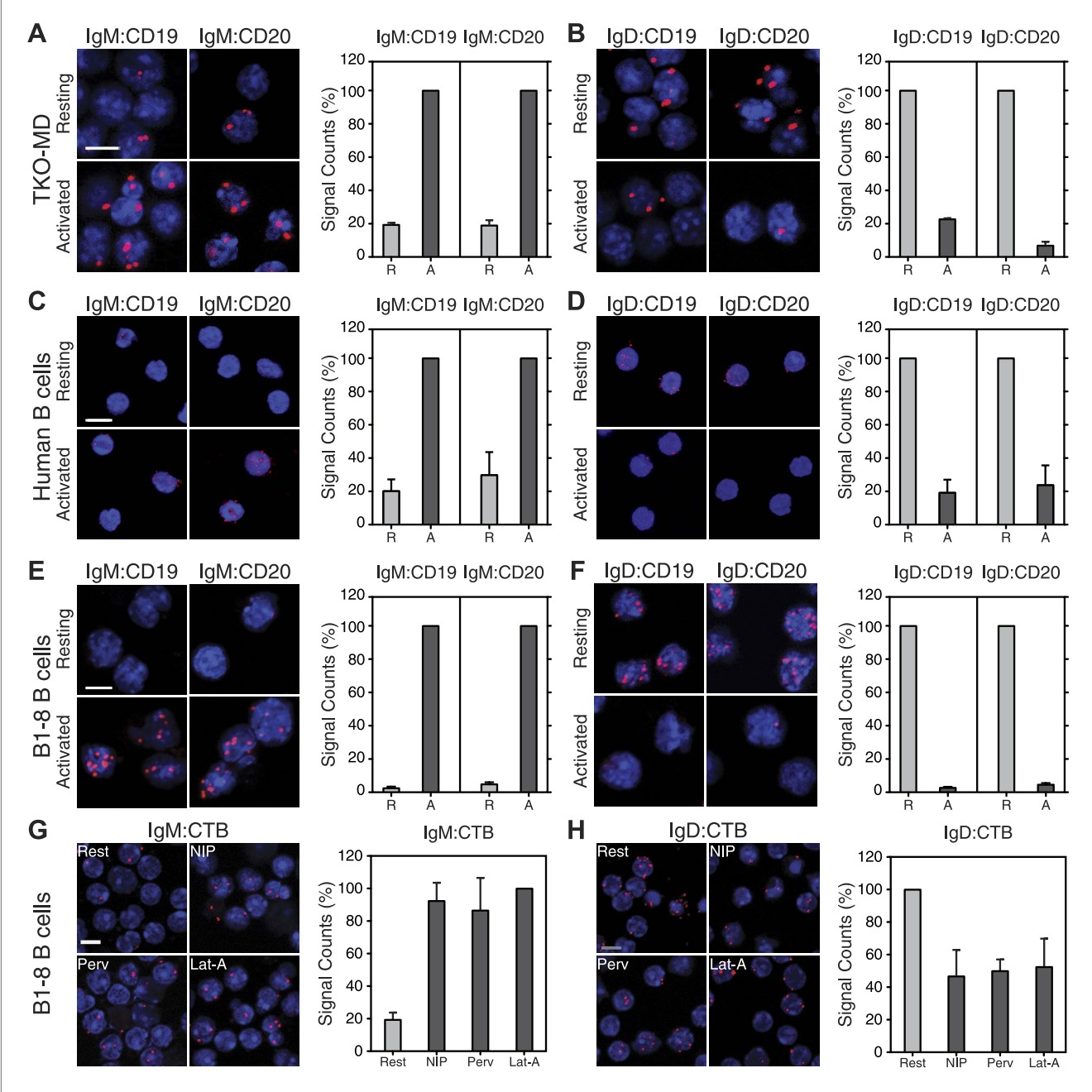

**Figure 7**. Fab-PLA of the nanoscale proximities of the IgM-BCR and IgD-BCR to the coreceptor molecules CD19/CD20 and the GM-1 Ganglioside. Representative microscopic images (left) and quantified results (right) of the proximity of CD19 or CD20 to the IgM-BCR (**A**, **C**, **E**) or the IgD-BCR(**B**, **D**, **F**) on resting or activated (**A** and **B**) TKO-MD, (**C** and **D**) human B cells and (**E** and **F**) B1-8 B cells stimulated for 5 min with NIP-BSA. (**G** and **H**) The proximity of the GM-1 Ganglioside (detected by CTB) to (**G**) the IgM-BCR or (**H**) the IgD-BCR on resting or activated B1-8 B cells stimulated for 5 min with the antigen NIP-BSA, the oxidant pervanadate and the F-actin inhibitor Lat-A, are shown as representative microscopic images (left) and quantified results (right). Scale bar: 5 μm. For the quantified results, data represent the mean and SEM of a minimum of three independent experiments. For each experiment, PLA signals (counts per cell) of each sample were counted from a minimum of 250 cells and were then normalized to the PLA signals of either (in **A**, **C**, **E**) the activated, or (in **B**, **D**, **F**, **H**) the resting B cells, or in (**G**) the Lat-A activated cells.

The following figure supplements are available for figure 7:

**Figure supplement 1**. Nanoscale IgM-BCR and IgD-BCR dissociation and coreceptor reorganization on B cells expressing only one BCR isotype.

**Figure supplement 2**. Fab-PLA of the nanoscale proximities of the IgM-BCR and IgD-BCR to the lipid raft marker CD52.

B cell lines or normal B cells of either mouse or human origin, using three different stimulation protocols and different sets of Fab fragments (*Figure 4*).

Previous live cell imaging studies of the behavior of the BCR on activated B lymphocytes have detected the formation of higher organized structures such as receptor microclusters and immunological synapses (*Treanor and Batista, 2007*; *Tolar et al., 2008*; *Harwood and Batista, 2011*). These studies have suggested that receptor aggregation rather than dissociation is the earliest event in B cell activation. However, due to the diffraction barrier of visible light of 250 nm (*Huang et al., 2010*), these live cell imaging studies could never directly monitor the conformation and nanoscale organization of the BCR on resting or activated B cells. In contrast to the expectations, several studies have now shown that antigen receptors on resting lymphocytes are not uniformly distributed monomers but rather pre-organized in nanoclusters or protein islands. The existence of these structures was first indicated by electron microscopy studies (*Wilson et al., 2000*) and recently confirmed by photo-activated localization microscopy (PALM) (*Lillemeier et al., 2006*, *2010*; *Sherman et al., 2011*) and direct stochastic optical reconstruction microscopy (dSTORM) (*Mattila et al., 2013*; *Owen et al., 2013*). In contrast to the BCR which is predominantly oligomeric, the T cell antigen receptor (TCR) seems to exist in an equilibrium between monomeric and oligomeric forms (*Schamel et al., 2005*). This may be the reason why TCR nanoclusters are more variable in size than the BCR nanoclusters (*Lillemeier et al., 2010*; *Sherman et al., 2011*; *Mattila et al., 2013*). However, common to the two cell types is the finding that upon activation the nanoclusters can form microclusters or cap structures (*Lillemeier et al., 2010*). Receptor dissociation and aggregation are thus not mutually exclusive events but rather processes that occur at different size levels and time points during B cell activation. Nanoclusters containing opened BCR presumably aggregate to form the microclusters previously detected by live cell imaging studies of activated lymphocytes.

By activating the PI-3 kinase signaling module, the CD19 coreceptor, together with the B cell adaptor for PI3K (BCAP), plays an important role as an amplifier of the BCR signal in activated B cells (*Aiba et al., 2008*). Indeed, CD19-deficient mice are defective in B cell development and function (*Rickert et al., 1995*; *Sato et al., 1997*). It has been shown that CD19 and IgM co-cap on activated B cells (*Pesando et al., 1989*) and it thus was no surprise to find the two receptors in close proximity to each other on these cells. Remarkably, however, we found that on resting B cells, CD19 and CD20 are in close proximity to the IgD-BCR but not the IgM-BCR. The resting IgD-BCR seems to be also associated with lipid ordered domains, as indicated by the proximity of this receptor to GM1 gangliosides and the GPI-linked protein CD52. The co-localization of the IgM-BCR with CD19 and lipid ordered domains is a hallmark of B cell activation but this is apparently not the case for the IgD-BCR on resting B lymphocytes. Thus, the IgD-BCR protein islands are likely to contain inhibitory factors which prevent signaling and we are currently searching for such molecules by Fab-PLA. These studies may help to elucidate the still enigmatic function of IgD on the B cell surface (*Geisberger et al., 2006*).

The finding that Syk is not only a downstream signaling element of the BCR, but directly involved in the opening of the oligomeric BCR, is one of the most surprising results of our nanoscale receptor studies. This discovery was only possible because Fab-PLA allows us, for the first time, to specifically monitor the oligomeric organization of the IgM molecule either in solution (*Figure 2*) or on the B cell surface. Our results indicate that Syk opens the BCR by an inside-out-signaling mechanism and makes B cells highly sensitive to antigen engagement by a feed-forward amplification process. An inside-out-signaling mechanism has previously been found to play a role in the activation of CD28 by the TCR (*Sanchez-Lockhart et al., 2011*) and the regulation of integrin receptors such as LFA-1 (*Wang, 2012*). In the latter case it is the binding of talin to the cytosolic tail of the β-chain that displaces the α-tail from its complex with the β-tail, thus moving the two tails apart. This intracellular movement is translated into an extracellular conformational change that opens the integrin for ligand binding. Similarly, the binding of Syk to the phosphorylated ITAM sequences may distort the Igα/Igβ tails in such a way that it destabilizes the BCR oligomer, thus resulting in BCR opening and activation. In line with this scenario, we found that a mutation of the disulfide bridge between Igα and Igβ, in combination with a transmembrane mutation of mIgD, prevents the formation of the IgD-BCR oligomer (*Yang and Reth, 2010b*).

For antibody production to occur in infected or immunized animals, the antigen must bind to the B cell carrying the cognate BCR. With the inside-out Syk/BCR signaling process described here only a few antigen/BCR/Syk complexes can activate all antigen receptors on the B cell surface, thus leading to signal amplification and full B cell activation (*Figure 5H*). This mechanism may allow B cells to become active even under limited antigen conditions. Our findings on the BCR/Syk signal amplification could be important for a better understanding of several human diseases. Recent data show that BCR

components are frequently mutated in human leukemia (*Davis et al., 2010*) and that an autonomously signaling BCR can act as a tumor promoter for B cell chronic lymphocytic leukemia (*Duhren-von Minden et al., 2012*). Clearly, it is important to learn more about the nanoscale BCR organization and its regulation in different disease settings.

## Materials and methods

### Antibodies

For FACS analysis of mouse spleen B cells or transfected TKO cells, following fluorophore- conjugated anti-mouse antibodies were used: Anti-CD45R-PerCP-Cy5.5 (RA3-6B2), Anti-IgM-APC, IgM-PE, IgM-FITC (II/41; all eBioscience, Frankfurt, Germany), Anti-CD20-APC, Anti-CD19-PE-Cy7, anti-IgD-AF647, Anti-IgD-APC, Anti-IgD-PE (11-26c.2a; all eBioscience).

For PLA probes against specific targets, the following unlabelled antibodies were used: IgD (11-26c.2a, SouthernBiotech, Birmingham, AL), IgD (AMS9.1; Santa Cruz Biotechnology, Dallas, TX), IgM (R33.24.12, in house hybridoma culture), IgM (rabbit anti-mouse μHC; Rockland Immunochemicals, Gilbertsville, PA), IgM (1B4B1; SouthernBiotech), Lambda light chain (JC5-1; SouthernBiotech), Kappa light chain (187.1; SouthernBiotech), CD19 (6D5; AbD Serotec, Düsseldorf, Germany) and CD20 (AISB12; eBioscience). Igα (HMK7/A9; abcam, Cambridge, UK), Syk (Syk-01; BioLegend, San Diego, CA).

For PLA probes against human BCR, the following unlabelled antibodies were used: IgD (IA6-2; BioLegend), IgD (IADB6) and IgM (SA-DA4) from Acris Antibodies (Herford, Germany), and IgM (Fc5u) from Genway Biotech (San Diego, CA).

### Cell lines, cell culture and transfection

The triple deficient pro B cell line (Rag2$^{-/-}$, λ5$^{-/-}$, SLP65$^{-/-}$) (TKO, *Meixlsperger et al., 2007*) is a kind gift of Hassan Jumaa. To generate the TKO-MD cells, TKO cells were retrovirally co-transfected with vectors encoding the λ1 light chain, B1-8 μm and δm HC carrying selection markers for puromycin and complemented YFP (*Köhler et al., 2008*; *Infantino et al., 2010*). The resulting cells were stained with 1NIP-peptide-DyLight649 (custom synthesized from Biosyntan GmbH, Berlin, Germany) for surface NIP-specific BCR and sorted for cYFP+ and Dylight649+ on BD (San Jose, CA) Influx FACS sorter.

To transfect TKO cells, supernatants containing viral particles were collected from transfected Phoenix retrovirus packaging cells 2 days after plasmid transfection and used as described before (*Duhren-von Minden et al., 2012*).

Both the TKO cells and the TKO-MD cells were cultured in Iscove's medium (Biochrom, Berlin, Germany) supplemented with 10 mM L-glutamine, 100 unit/ml penicillin/streptomycin (all Gibco, Life Technologies, Darmstadt, Germany), 50 μM β-mercaptoethanol (Sigma-Aldrich, Munich, Germany), 10% FCS (PAN Biotech, Aidenbach, Germany) and supernatant of cultured J558L mouse plasmacytoma cells stably transfected with a murine IL-7 expression vector.

### Isolation of mouse spleen B cells

Total spleen cells were isolated from 8–12 week old B1-8 transgenic mice harboring NIP-specific B cells. Naïve B cells were enriched by MACS depletion method using anti-CD43 magnetic beads (Miltyeni Biotech, Bergisch Gladbach, Germany) according to manufacturer's protocol. Before their stimulation, the enriched B cells were rested overnight in complete Iscove's medium as described above and their activation status was monitored by flow cytometry analysis after staining with an anti-CD86 antibody.

### Isolation of human naïve B cells

Total PBMCs (peripheral blood mononuclear cells) were prepared from freshly withdrawn anti-coagulated peripheral blood about 15–20 ml from a healthy donor using Leucosep 50 tube (Greiner Bio One, Frickenhausen, Germany) and Pancoll, 1.077 g/l (PAN Biotech) according to manufacturers' instruction. PBMC washed with PBS and naïve B cells fraction was enriched by MACS depletion method using Naive B cell isolation kit II (Miltyeni Biotech). To verify the purity and identify subpopulations such as immature, plasma and memory B cells, isolated fractions were analyzed in FACScan using different B cell surface markers that includes IgD, IgM, CD10, CD19, CD20, CD27, CD43. Thereafter these B cells, prior to PLA experiment, were cultured at 37°C in 5% CO2, in RPMI 1640 media supplemented with 25 mM Hepes, and 10 units/ml penicillin/streptomycin (all from Invitrogen, Life Technologies, Karlsruhe, Germany) and 2% of filter sterilized self plasma (separated during PBMC preparation) for 8–12 hr to rescue from the stress occurred during isolation.

## Cell activation and kinase inhibition

To activate the BCR, cells were treated with 1 μM Latrunculin A (Invitrogen), 0.5 mM of pervanadate, or 40 ng/ml NIP-BSA (Biosearch, Petaluma, CA) for the indicated times. Pervanadate was freshly prepared for each experiment with equal molar amounts of orthovanadate and $H_2O_2$. For kinase inhibition, cells were pre-treated with 5 μM Syk inhibitor R406 (Selleckchem, Houston, TX) or 10 μM Lyn inhibitor PP2 (Sigma-Aldrich) for 60 min before the stimulation.

## Fab-PLA probes preparation

Fab fragments were prepared from the corresponding antibodies with Pierce-Fab-Micro preparation kit (Thermo Fisher Scientific, Bonn, Germany) using immobilized Papain or Ficin according to manufacturer's protocol. After desalting (Zeba spin desalting columns, Thermo Fisher Scientific), the resulted Fab-fragments were coupled with PLA-probemaker plus or minus oligonucleotides according to the manufacturer's instructions (Olink Bioscience, Uppsala, Sweden) to generate Fab-PLA probes.

## PLA protocol

For in situ PLA, cells were settled on PTFE-slides (Thermo Fisher Scientific) for 30 min at 37°C. Resting and activated cells were fixed for 15 min with 2% paraformaldehyde containing 0.02% glutaraldehyde in PBS for faster crosslinking at room temperature. Reduction of the aldehyde groups was achieved by incubation for 10 min with 0.5 mg/ml NaBH₄. For intracellular PLA, cells were permeabilized after fixation with 0.5% saponin (quillaja bark, Sigma) in PBS for 30 min. PLA reactions were performed based on a previously described protocol (*Soderberg et al., 2008*). Briefly, blocking solution contains 25 μg/ml sonicated salmon sperm DNA and 250 μg/ml BSA in PBS. Any treatment with EDTA or tween was avoided. After blocking, cells were incubated with appropriate PLA probes for Fab-PLA, 1-PLA or with primary antibody for 2-PLA in probemaker diluent. In the case of 2-PLA, cells received further incubation with secondary PLA probes Duolink II kit (Olink Biosience) binding to the corresponding primary antibody. PLA signal amplification was performed following manufacturer's instruction. Resulting samples were directly mounted on slides with DAPI-Fluoromount-G (Southern Biotech) to visualize the PLA signals in relation to the nuclei.

## Detecting IgM:IgM proximity of pentameric and monomeric IgM bound to latex beads

Carboxylate modified latex (CML) beads with the diameter of 10 μm (Invitrogen) were coupled with NIP(15)BSA-biotin (Biosearch) following manufacturer's instruction. Briefly, 250 μl of CML beads (40 mg/ml) were first washed twice with 1 ml MES buffer (25 mM, pH6) and then resuspended in 500 μl MES. 200 μl of freshly prepared EDAC (1 Ethyl 3-(3-Dimethyl Amino Propyl) Carbodiimide HCl, Sigma-Aldrich, 50 mg/ml in MES) and 300 μl NIP(15)BSA-biotin (Biosearch, 6.6 μg/ml in MES) were added to the latex suspension and incubated at room temperature with gentle mixing for 4 hr. The resulting beads were then washed three times with 1 ml PBS and resuspended in 1 ml storage buffer (1X PBS with 0.1% glycine and 0.1% NaN3).

Purified monomeric and pentameric IgM were loaded on beads by incubating 100,000 beads with 0.05 μM (monomeric) or 0.01 μM (pentameric) of IgM in 30 μl volume at room temperature for 30 min under mild shaking. The equal loading of the monomeric and pentameric IgM was verified by staining the beads with anti-IgM-APC (eBioscience) antibody and monitoring the staining by FACScan. After loading, the beads were washed with PBS and PLA were performed as described. The PLA signal was quantified by a FACScan on LSRFortessa (Becton Dickinson, Franklin Lakes, NJ) using the filter set for PE. Data were exported and analyzed and plotted using FlowJo software (TreeStar, Ashland, OR).

## Imaging and image analysis

All microscopic images were acquired on a Zeiss 780 Meta confocal microscope (Carl Zeiss, Jena, Germany) using a Zeiss Plan-Apochromat 63X oil immersion objective lens. For each sample, a minimal of five images were captured from randomly chosen regions. All recorded images were analyzed by single cell analysis using the BlobFinder (Centre for Image Analysis, Uppsala university) software.

## Data processing and statistical analysis

Raw data produced by BlobFinder were exported to Prism (Graphpad, La Jolla, CA) software. For each sample in each experiment, the average PLA signal counts per cell was calculated from the

corresponding images and then normalized to the average PLA signal counts per cell of the reference sample in the same experiment. These normalized PLA signal counts from a minimum of three independent experiments were used for the plot and to calculate the difference between samples. A one-tailed paired $t$ test was used to determine the p value.

## Flow cytometry analysis of Fab-PLA probe binding

Fab-PLA probes were labeled by annealing with fluorescence-coupled complementary oligonucleotides and size separated to remove excess unbound oligos. Resting and activated B1-8 cells were fixed for 15 min with 2% paraformaldehyde in PBS at room temperature. Thereafter, the fixed cells were incubated with fluorescence labeled Fab-PLA probes in blocking solution containing 250 µg/ml BSA, 2.5 µg/ml sonicated salmon sperm DNA, washed with PBS and subjected to flow cytometry analysis using a FACScan instrument. Resting cells treated with matching concentration of dsDNA prepared by annealing free plus or minus oligo with the corresponding fluorescence coupled complementary oligo were used as a control.

## Schneider cell culture and transient transfection

Schneider S2 cells were cultured and transfected as described previously (*Yang and Reth, 2012*). To induce the protein expression of the transfected plasmids, cells were treated with 1 mM $CuSO_4$ for 24hr. Cells were co-transfected with plasmids encoding BCR and GFP tagged Syk (wt or mutant) were sorted for GFP-expression. Cells without the co-transfection of Syk were stained by anti-λ-FITC and FITC-positive cells were purified by cell sorting.

## Acknowledgements

We thank Peter Nielsen, Aaron Marshall, Hassan Jumaa and Wolfgang Schamel for critical reading of this manuscript. We thank Hassan Jumaa for the TKO pro B cell line, Pavel Salavei for purifying monomeric and pentameric IgM and Christa Kalmbach-Zürn for S2 cells. We also thank Klaus Rajewsky and Sacha Tarakovsky for the B1-8 and Syk[fl/fl] mice, respectively. This study was supported by the Excellence Initiative of the German Federal and State Governments (EXC294), by ERC-grant 322972 and by the Deutsche Forschungsgemeinschaft through SFB746 and TRR130.

## Additional information

### Funding

| Funder | Grant reference number | Author |
| --- | --- | --- |
| Deutsche Forschungsgemeinschaft (DFG) | Excellence Initiative of the German Federal and State Government, EXC294 | Michael Reth |
| European Research Council (ERC) | Advanced Grant, 322972 | Michael Reth |
| Deutsche Forschungsgemeinschaft (DFG) | SFB746 | Michael Reth |
| Deutsche Forschungsgemeinschaft (DFG) | TRR130 | Michael Reth |

The funder had no role in study design, data collection and interpretation, or the decision to submit the work for publication.

### Author contributions

KK, Developed the Fab-PLA and conducted the experiments; PCM, Developed the Fab-PLA and conducted some of the experiments; EH, Generated the mice allowing the deletion of the Syk gene in mature B cells; JY, Planned the experiments. Helped prepare the manuscript; MR, Planned the experiments. Prepared the Manuscript.

### Ethics

Animal experimentation: Experiments with animals were reviewed by the institutional animal ethics committee and were performed according these approved procedures (Permit Re-TO5).

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
