## [Decision Letter]

Thank you for sending your work entitled “B cell activations involve nanoscale receptor reorganizations and inside-out signaling by Syk” for consideration at *eLife*. Your article has been favorably evaluated by Tony Hunter and 3 expert reviewers.

The Senior editor and the reviewers discussed their comments before a consensus decision was reached, and the Senior editor has assembled the following comments to help you prepare a revised submission.

There is general agreement that your findings are potentially an important advance in understanding the mechanism of BCR signal transduction, but that more needs to be done to bolster the main conclusion that BCR-antigen interaction induces dimer dissociation in a manner dependent on Syk activity. In particular, there are concerns about the interpretation of the PLA data and how well the PLA analysis measures distance between the two BCR subunits, and whether your results mean that the BCR dimers actually dissociate or rather reorient and/or undergo a conformational rearrangement. The main issues that would have to be addressed before the paper could be accepted by *eLife* are:

1) The authors use three types of PLA reagents coupling oligos to secondary Abs (2-PLA), primary Abs (1-PLA) or to Ab Fabs (Fab-PLA), and the reported resolution power of these methods is based solely on what appears to be a theoretical detection distance range based on the estimated end to end 'size' of an antibody, and from this the authors deduce that the different PLA assays detect BCRs at various distances from each other. Clearly, they need to demonstrate that the PLA assays are indeed sensitive to the distances they theorize, for example, by creating artificial surfaces displaying BCRs at these various distances or making artificial (covalent) dimers. In this regard, do the authors know the stoichiometry of binding of the PLAs to the cells? Does the μ-specific mAb bind to both IgH chains in the BCR? If so does this contribute to signal?

2) The PLA results indicate that the distance between the constant domains of Igμ increases after stimulation but not necessarily that BCR dimers dissociate upon antigen binding. The use of an independent approach to validate the PLA results and determine whether true dissociation actually occurs could strengthen the conclusions. Ideally, the authors should back-up their PLA data with data generated by a completely different technique, such as EM or native gels. If results could be obtained using another method in a reasonable time frame, then these should be added. In addition, the use of BiFC to show that the Syk inhibitor also blocks the “opening” of the oligomer measured by this technique would be reassuring. In summary, unless they can provide incontrovertible evidence that dissociation occurs, the word dissociation should be replaced with “opening” or an equivalent term.

3) The authors refer to the observed change in BCR subunit proximity as dissociation of the dimer; however, their data indicate that the distance between the cytoplasmic domains increases from the 10-20 nm distance detected by their Fab probe (Fab-PLA) to between 20-40 nm detected with their primary Ab probe (1-PLA). This points more to an opening of the receptor dimer, as the domains are now less 40 nm apart, rather than a complete dissociation. How would the authors interpret a conformational change rather than dissociation in terms of the mechanism of BCR activation?

In the discussion of Syk-induced activation the authors alternate between describing events as receptor opening or dissociation as if the two words mean the same thing. They should be more precise in how much of a separation they believe is occurring during activation and how this leads to increased sensitivity. Indeed, the cartoon shown in Figure 6 indicates that they believe the BCR dimers are completely separate and free to diffuse. They need to explain how they have interpreted the 1-PLA results and how those results support complete dissociation or they could include a model showing dimer opening.

4) As the authors describe their analysis, the PLA assay samples only a few BCR interactions per cell (∼5-10) and these data are used to draw conclusions about the behavior of a larger pool of BCRs. Is this extrapolation justified for B cells? It is well established that BCR surfaces are not uniform and that ruffles and ridges can alter local concentrations of receptors. Moreover, upon activation such membrane structures can change dramatically. The authors could be measuring BCR clusters in ridges or ruffles that are not present throughout the membrane of activated cells. Indeed, the drugs used to modify BCR clustering behavior (i.e. latrunculin, pervanadate) alter cell morphology. The authors need to address these issues including the use of a membrane dye to determine if the observed PLA signal is selectively from areas of deformation of the membrane.

5) The authors quantify the binding of the PLAs to resting and activated cells to make the point that Abs don't dissociate upon BCR activation. But if only small numbers of BCR clusters are sampled, loss of Ab binding would not be detectable by flow analysis. It is also necessary to verify the technique using PLAs specific for other regions of the BCR, i.e. Igα/Igβ, L chain or Fab.

6) Based on the use of a Syk kinase inhibitor and Syk overexpression, the authors conclude that Syk ITAM phosphorylation and binding of Syk to the phosphorylated Igα/Igβ subunits is required for BCR dissociation/opening. However, the results could alternatively be explained by steric inhibition of oligomerization through Syk binding or even simple repulsion due to the negative charge created by ITAM phosphorylation. They need to test the requirement for Syk binding through expression of the Syk kinase domain without the tandem SH2 domains or Syk with mutated SH2 domains, expression of tandem SH2 and kinase domains as separate constructs, or mutation of the ITAM motifs in the S2 system.

7) The discussion of prior literature regarding clustering of antigen receptors is rather selective and does not give a balanced view. For instance, the authors state “recent studies show that antigen receptors on resting T and B cells are not uniformly distributed but are rather organized in nanoclusters inside protein islands (22, 21, 23)”. In fact, there is disagreement over the size of T cell nanoclusters and whether they form a continuous distribution or islands. Indeed, they fail to cite Sherman et al. (Immunity 35, 705, 2011) who also used photoactivated colocalization microscopy to assess TCR distribution but did not observe the large nanoclusters reported by Lillemeier et al. They mention several studies that show T cell clusters range between 30 and 300 nm, but fail to cite studies that support smaller cluster sizes (A native gel paper (Schamel et al. J Exp Med 202:493, 2005), STORM data showing a continuous size distribution (Owen et al. J Biophotonics 3:446, 2010), as well as a PALM paper showing small nanoclusters (Sherman et al. op.cit.). Although the authors claim that the size of TCR clusters and the role of concatenated islands in T cell activation is established doctrine, in fact it remains a hypothesis and should be explained as such in the discussion, which needs to be much more balanced.

---

## [Author Response]

1) The authors use three types of PLA reagents coupling oligos to secondary Abs (2-PLA), primary Abs (1-PLA) or to Ab Fabs (Fab-PLA), and the reported resolution power of these methods is based solely on what appears to be a theoretical detection distance range based on the estimated end to end 'size' of an antibody, and from this the authors deduce that the different PLA assays detect BCRs at various distances from each other. Clearly, they need to demonstrate that the PLA assays are indeed sensitive to the distances they theorize, for example, by creating artificial surfaces displaying BCRs at these various distances or making artificial (covalent) dimers. In this regard, do the authors know the stoichiometry of binding of the PLAs to the cells? Does the μ-specific mAb bind to both IgH chains in the BCR? If so does this contribute to signal?

As the reviewers asked us to provide further evidence that our Fab-PLA technique is indeed sensitive to the epitope distance, we searched, as suggested, for artificial molecules with precise epitope spacing to test this method. It turns out that for PLA we have to place such molecules on a surface in a way that we cannot properly control their proximity. It is thus very difficult to create a uniform surface with defined epitope spacing. However, by thinking about this we came up with an even better solution for this problem; instead of testing artificial surfaces or dimers we studied a natural molecule existing as either monomer or oligomer namely soluble IgM. We thus purified monomeric and pentameric IgM and bound limiting amounts of them to hapten-coupled latex beads. After controlling equal loading of the beads we conducted Fab-PLA and 1-PLA with the monomeric or pentameric IgM containing beads. The results were clear-cut. Our Fab-PLA method could only detect the pentameric but not the monomeric IgM, whereas 1-PLA detected both IgM forms equally well (see new Figure 2). This is a direct proof that only Fab-PLA but no other method is specifically detecting the oligomeric BCR organization. In the first version of our manuscript we came to the same conclusion by studying a wild type or a double mutant IgD-BCR unable to form oligomers (Figure 3). We thus think that Fab-PLA is the method of choice to monitor nanoscale alterations in receptor organisation. Although, we do not know whether the used μ-specific mAb has two or (more likely) only one epitope per monomeric IgM this is not relevant for our method as we now have shown that Fab-PLA does not detect isolated IgM monomers unless they are spaced in close proximity to each other.

*2) The PLA results indicate that the distance between the constant domains of Igμ increases after stimulation but not necessarily that BCR dimers dissociate upon antigen binding. The use of an independent approach to validate the PLA results and determine whether true dissociation actually occurs could strengthen the conclusions. Ideally, the authors should back-up their PLA data with data generated by a completely different technique, such as EM or native gels. If results could be obtained using another method in a reasonable time frame, then these should be added. In addition, the use of BiFC to show that the Syk inhibitor also blocks the “opening” of the oligomer measured by this technique would be reassuring. In summary, unless they can provide incontrovertible evidence that dissociation occurs, the word dissociation should be replaced with “opening” or an equivalent term*.

The reviewers asked us to use additional independent approaches to validate our Fab-PLA experiments and prove that indeed mIg dissociates during BCR activation. Unfortunately, apart from Fab-PLA there is presently no other method, be it superresolution or EM, that can reliably distinguish between a closed and opened BCR oligomer with alterations in the 10-20 nm distance. This is what makes our Fab-PLA assay so special and exiting and we think this approach can be used to address issues related to receptor organization in membrane biology and will be of interest for the broad readership of *eLife.* The referee suggests using the bifluorescence complementation assay (BiFC) in combination to Syk inhibition to study the requirements for BCR dissociation. Unfortunately, this is not possible. Once the fluorophore has formed the two halves of GFP are linked to each other and cannot be dissociated in living cells (Hu et al. 2002). We also want to point out that, whereas BCR opening can so far only reliably be studied by Fab-PLA, many different methods (blue native gels, BiFC, EM and superresolution) were used to provide evidence for the clustering and oligomeric organization of the BCR ([36]; Yang & Reth 2010; [23]; Fiala et al. 2013).

3) The authors refer to the observed change in BCR subunit proximity as dissociation of the dimer; however, their data indicate that the distance between the cytoplasmic domains increases from the 10-20 nm distance detected by their Fab probe (Fab-PLA) to between 20-40 nm detected with their primary Ab probe (1-PLA). This points more to an opening of the receptor dimer, as the domains are now less 40 nm apart, rather than a complete dissociation. How would the authors interpret a conformational change rather than dissociation in terms of the mechanism of BCR activation?

*In the discussion of Syk-induced activation the authors alternate between describing events as receptor opening or dissociation as if the two words mean the same thing. They should be more precise in how much of a separation they believe is occurring during activation and how this leads to increased sensitivity. Indeed, the cartoon shown in*
Figure 6
*indicates that they believe the BCR dimers are completely separate and free to diffuse. They need to explain how they have interpreted the 1-PLA results and how those results support complete dissociation or they could include a model showing dimer opening*.

The reviewers suggest that we shall replace the term “dissociation” with the term “opening”. We agree with the referees that our Fab-PLA method cannot distinguish between a complete dissociation or an opening of the BCR and we therefore now use the term “opening” throughout the new version of our manuscript. We want to point out, however, that these two terms do no exclude each other. Although it is possible that after BCR activation “opened” mIg:mIg dimers are still held together for example by Syk the mIg transmembrane parts could become dissociated from each other and this maybe the reason for the changes of the lipid environment around the activated BCR that we observe in our IgM:CTB PLA study (Figure 7).

*4) As the authors describe their analysis, the PLA assay samples only a few BCR interactions per cell (∼5-10) and these data are used to draw conclusions about the behavior of a larger pool of BCRs. Is this extrapolation justified for B cells? It is well established that BCR surfaces are not uniform and that ruffles and ridges can alter local concentrations of receptors. Moreover, upon activation such membrane structures can change dramatically. The authors could be measuring BCR clusters in ridges or ruffles that are not present throughout the membrane of activated cells. Indeed, the drugs used to modify BCR clustering behavior (i.e. latrunculin, pervanadate) alter cell morphology. The authors need to address these issues including the use of a membrane dye to determine if the observed PLA signal is selectively from areas of deformation of the membrane*.

The reviewers rightly point out that the cellular membrane is not uniform but also contains ruffles and ridges that can alter in the local concentration of receptors and that these structures can change after activation. However, we could not find any evidence in the literature to support the assumption that BCR oligomers are special structures existing only in membranes with a specific topology. In our BiFC studies we found that BCR dimers are equally distributed on the S2 Schneider cell surface and not restricted to specialized membrane structures (Yang & Reth 2010). We also showed that a mutant BCR that cannot form a BCR oligomer is hardly expressed on the cell surface. We therefore think that the majority of the BCR on the cell surface form oligomers.

*5) The authors quantify the binding of the PLAs to resting and activated cells to make the point that Abs don't dissociate upon BCR activation. But if only small numbers of BCR clusters are sampled, loss of Ab binding would not be detectable by flow analysis. It is also necessary to verify the technique using PLAs specific for other regions of the BCR, i.e. Igα/Igβ, L chain or Fab*.

The reviewers suggest that we also do other PLA techniques to prove that the alterations in Fab-PLA between resting and activated B cells are not due to a loss of antibody binding, We have controlled that point in our FACS scan analysis (Figure 2—figure supplement 1). However, we also provided data in the manuscript showing Fab-PLA between the heavy and light chain that remains unchanged during the activation process (Figure 2—figure supplement 1) and we think that we have thus taken care of this point.

*6) Based on the use of a Syk kinase inhibitor and Syk overexpression, the authors conclude that Syk ITAM phosphorylation and binding of Syk to the phosphorylated Igα/Igβ subunits is required for BCR dissociation/opening. However, the results could alternatively be explained by steric inhibition of oligomerization through Syk binding or even simple repulsion due to the negative charge created by ITAM phosphorylation. They need to test the requirement for Syk binding through expression of the Syk kinase domain without the tandem SH2 domains or Syk with mutated SH2 domains, expression of tandem SH2 and kinase domains as separate constructs, or mutation of the ITAM motifs in the S2 system*.

The reviewers rightly pointed out that in our S2 BCR rebuilding studies showing that Syk is opening the BCR oligomer we have not determined whether it is the ITAM phosphorylation or the binding of the receptor by Syk that results in BCR opening. We therefore have now conducted an S2 experiment separating the enzymatic and binding function of Syk from each other (new Figure 6). Specifically we co-expressed the IgM-BCR and a membrane-bound Syk kinase domain either alone or in combination with a GFP-(SH2)2 fusion protein containing the tandem SH2 domains of Syk. The result of this study was clear-cut. Membrane-bound Syk is phosphorylating the ITAM tyrosines of Igα but this does not result in BCR opening as measured by Fab-PLA. It is only the combination between the membrane-bound kinase domain and the GFP-(SH2)2 construct that results in BCR opening, showing that it is the binding of Syk and not the phosphorylation that initiates the opening process. This notion is also supported by our older data showing that the inhibition of the kinase activity of Syk prevents not only receptor opening but also the binding of Syk to the BCR (Figure 5). We thus provide in the new version of our manuscript a clear-cut answer to this question and more information concerning the opening process of the oligomeric BCR.

*7) The discussion of prior literature regarding clustering of antigen receptors is rather selective and does not give a balanced view. For instance, the authors state “recent studies show that antigen receptors on resting T and B cells are not uniformly distributed but are rather organized in nanoclusters inside protein islands (*[22]*,*
[21]*,*
[23]*)”. In fact, there is disagreement over the size of T cell nanoclusters and whether they form a continuous distribution or islands. Indeed, they fail to cite Sherman et al. (Immunity 35, 705, 2011) who also used photoactivated colocalization microscopy to assess TCR distribution but did not observe the large nanoclusters reported by Lillemeier et al. They mention several studies that show T cell clusters range between 30 and 300 nm, but fail to cite studies that support smaller cluster sizes (A native gel paper (Schamel et al. J Exp Med 202:493, 2005), STORM data showing a continuous size distribution (Owen et al. J Biophotonics 3:446, 2010), as well as a PALM paper showing small nanoclusters (Sherman et al. op.cit.). Although the authors claim that the size of TCR clusters and the role of concatenated islands in T cell activation is established doctrine, in fact it remains a hypothesis and should be explained as such in the discussion, which needs to be much more balanced*.

As suggested we have now mentioned this literature in our manuscript and changed the discussion of membrane organization on T and B cells.